

# Comparison of urine proteomes from tumor-bearing mice with those from tumor-resected mice

Ziqi Heng, Chenyang Zhao and Youhe Gao

Department of Biochemistry and Molecular Biology, College of Life Sciences, Beijing Normal University, Gene Engineering Drug and Biotechnology Beijing Key Laboratory, Beijing, China

## ABSTRACT

**Objective**. This study aimed to address on the most important concern of surgeons—whether to completely resect tumor. Urine can indicate early changes associated with physiological or pathophysiological processes. Based on these ideas, we conducted experiments to explore changes in the urine proteome between tumor-bearing mice and tumor-resected mice.

**Method**. The tumor-bearing mouse model was established with MC38 mouse colon cancer cells, and the mice were divided into the control group, tumor-resected group, and tumor-bearing group. Urine was collected 7 and 30 days after tumor resection. Liquid chromatography coupled with tandem mass spectrometry (LC–MS/MS) was used to identify the urine proteome, which was analyzed for differentially expressed proteins and functional annotation.

**Results**. (1) Seven days after tumor resection, 20 differentially expressed proteins distinguished the tumor-resected group and the tumor-bearing group. The identified biological processes included circadian rhythm, Notch signaling pathway, leukocyte cell–cell adhesion, and heterophilic cell–cell adhesion *via* plasma membrane cell adhesion molecules. (2) Thirty days after tumor resection, 33 differentially expressed proteins distinguished the tumor-resected group and the tumor-bearing group. The identified biological processes included cell adhesion; complement activation, the alternative pathway; the immune system process; and angiogenesis. (3) The difference in the urine proteome between the tumor-resected group and the healthy control group was smaller 30 days after tumor resection.

**Conclusion**. Changes in the urinary proteome can reflect the complete resection of MC38 tumors.

Corresponding author
Youhe Gao, gaoyouhe@bnu.edu.cn

## INTRODUCTION

The most common surgical treatment method for solid tumors is resection combined with chemotherapy or radiotherapy, but tumor recurrence after treatment is very common. Whether the tumor is completely resected is a major concern for many surgeons.

Every cell in the body depends on a stable internal environment to survive and function. Fluid in this environment provides cells with oxygen and nutrients and represents a conduit for excreting waste. The relative constancy of the internal environment is known

as homeostasis (*Rodriguez-Ruiz et al., 2017*). Blood, as the internal environment, must be stable and balanced to protect cells from disturbing factors. Changes do occur, but they are small and factors are kept within narrow limits. In contrast, urine, the fluid that is filtered from blood, does not need nor have a stabilizing mechanism and is not subject to homeostatic regulation. The urinary proteome is very rich in information. Some low-molecular weight proteins such as hormones and cytokines are quickly excreted into the urine after entering the blood; these proteins have a high probability of being detected in the urine (*Nagaraj & Mann, 2011*). Urine is able to enrich for changes caused by the body's disease in its early stage. Thus, urine is a good biological source of biomarkers (*Gao, 2013*).

Based on the above ideas, our laboratory has also carried out a series of studies. The urinary proteome of Walker 256 tumor-bearing rats showed significant changes prior to the formation of palpable tumor masses, and the factors that showed early changes in the urine also showed differential abundance in the late stages of cancer (*Wu, Guo & Gao, 2017*). Changes in the urinary proteome occurred on Day 2 after the tail vein injection of Walkers-256 cells in rats, earlier than the pathological changes in lung tumor nodules, which appeared on Day 4 (*Wei et al., 2019*). Twenty-five proteins in the urine of rats in the tumor group were significantly altered 3 days after Walker 256 cells were implanted in the tibia, preceding the detection of significant lesions by computed tomography (CT) (*Wang et al., 2020*). On Day 3 after the injection of Walker-256 cells into the liver, the levels of 12 proteins were significantly changed in the experimental rats, seven proteins were significantly associated with liver cancer, and the presence of the same type of tumor cells growing in different organs were reflected in the differential urine proteins (*Zhang, Gao & Gao, 2020*).

In other studies, researchers generated additional evidence to support urinary proteins as disease biomarkers. Comparative proteomics identified 21 increased and 8 decreased proteins among 870 identified urinary proteins in the mdx-4cv mouse model of dystrophinopathy. Nidogen, parvalbumin and titin were found almost exclusively in mdx-4cv mice (*Gargan et al., 2020*). *Song et al. (2020)* reported the differential expression of 88 proteins including 11 brain cell markers Annexin 2 and Clusterin enabling the detection of differences before amyloid-$\beta$-plaque deposition in the 5XFAD mouse model. *Zhang et al. (2015)* reported the significant upregulation of seven proteins in both Kras (G12D) mouse models. A recent study showed that the sensitivity of urinary diagnostic signature combined with FIT for CRC was improved compared with FIT alone, but also define a panel of four urinary biomarkers (CORO1C, RAD23B, GSPT5, and NDN) for CRC metastatic risk stratification for potential interventional targets (*Sun et al., 2022*). An approach that recently reported is based on the mass spectrometry (MS)-based proteomic analysis of urine samples from head and neck squamous cell carcinoma (HNSCC) and thyroid cancer patients, changes in the levels of 29 urinary proteins during and after therapeutic interventions were detected, which could serve as tumor biomarkers (*Ferrari et al., 2019*).

There are many factors affecting urine samples, and it is time-consuming to collect samples from patients with early-stage disease. Therefore, establishing a mouse model of

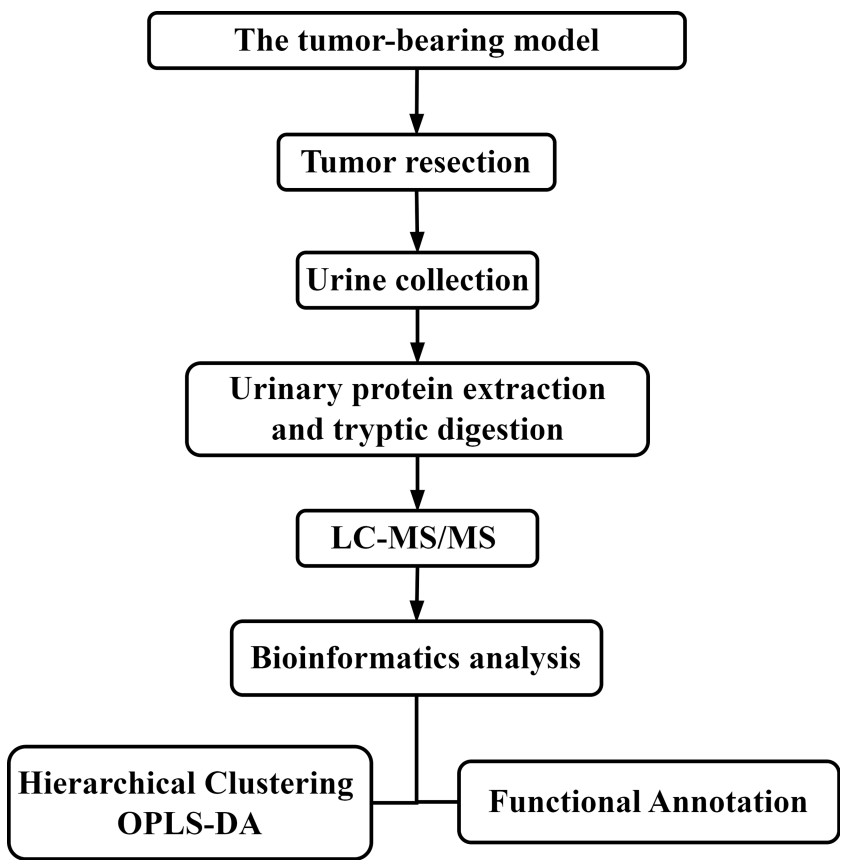

**Figure 1  The experimental design and workflow of the analysis in this study.** The tumor-bearing model was established, and then the tumors were resected from the MC38 tumor-bearing mice. Urine samples were collected at Days 7 and 30 after tumor resection. Urine proteins were identified by liquid chromatography coupled with tandem mass spectrometry (LC–MS/MS).

colorectal cancer minimizes potential interfering factors and allows the dynamic monitoring of disease progression and the collection of urine samples prior to pathological or clinical presentation, thereby facilitating research related to tumor resection and recurrence (*Wei & Gao, 2021*). In these initial stages of urine biomarker research, more comprehensive insight into disease is needed. Therefore, differentially expressed urinary proteins were selected using less stringent criteria and then subjected to functional enrichment analysis and comparison. In this study, subcutaneous tumors were generated using the MC38 cell line derived from a colon adenocarcinoma in a C57BL/6 mouse. Then, the tumors were resected from the tumor-bearing mice. Urine samples were collected after surgery, and liquid chromatography-tandem mass spectrometry (LC–MS/MS) was performed for urine proteomics analysis. The workflow is shown in Fig. 1. The aim of this study was to compare the urine proteomes of tumor-bearing mice with those of tumor-resected mice.

## MATERIALS & METHODS

### Establishment of the MC38 tumor-bearing mouse model

Male C57BL/6 mice (18 g–20 g) were supplied by Beijing Vital River Laboratory Animal Technology Co., Ltd. All animals were maintained with free access to standard laboratory diet and water with a 12-h light–dark cycle under constant temperature ($22 \pm 2\,°C$) and humidity (65–70%). The experiment was approved by the ethics and animal welfare committee of Beijing Normal University (Approval ID: CLS-EAW-2020-0344). All experiments were performed in accordance with relevant guidelines and regulations of the National Health Commission and the Ministry of Science and Technology. MC38 cells were obtained from the Cell Culture Center of Chinese Academy of Medical Sciences (Beijing, China) and were inoculated into C57BL/6 mice.

The experimental mice were acclimated in the new environment for three days. Then, they were randomly divided into three groups: a healthy control group ($n = 5$), a tumor-resected group ($n = 5$) and a tumor-bearing group ($n = 5$). MC38 cells were added to complete culture medium (DMEM supplemented with, 10% fetal bovine serum and 1% penicillin/streptomycin) and placed in T75 cell culture flasks. After a sufficient cell population was established, MC38 cells were collected, centrifuged, and resuspended in phosphate-buffered saline (PBS) for the subsequent establishment of the mouse models. Cell viability was assessed by the trypan blue exclusion test. Briefly, MC38 cells were stained with 0.4% trypan blue solution and then counted using a hemocytometer; tumor cell viability was greater than 95%.

The mice were anesthetized with 4 mg/kg sodium pentobarbital and placed on disposable sterile medical pads; then, hair was removed from the injection site, which was subsequently disinfected. The experimental mice in the tumor-resected group and tumor-bearing group received a subcutaneous injection of $5 \times 10^6$ viable MC38 cells in 200 µl of PBS into the right hind limb. The healthy control group was injected with PBS buffer at the same location. The mice in each of the three groups underwent resection after the establishment of the tumors. The subcutaneous tumor was completely resected from the mice in the tumor-resected group; In the tumor-bearing group, the subcutaneous tumor was preserved, and small section of the tissue and muscle were removed to ensure the same wound surface in all mice; In the healthy control group, small section of the tissue and muscle were removed in all mice to control variables in this study. After the experiment, all the animals were euthanized by an intraperitoneal injection of barbiturates.

### Urine sample collection and preparation

Urine was collected from mice in the three groups at Day 7 and 30 after resection. During urine collection, all mice were placed individually in metabolic cages with no food or water. Urine was collected overnight for 12 h, and the volume of urine collected was not less than 1 ml. The urine samples were stored at $-80\,°C$ immediately after collection.

Urine samples were defrosted at $4\,°C$ and centrifuged at $12,000 \times g$ for 30 min at $4\,°C$ to remove cell debris. Then, the supernatants were precipitated with three volumes of ethanol at $-20\,°C$, followed by centrifugation at $12,000 \times g$ for 30 min. The pellet was resuspended in lysis buffer (8 mol/L urea, 2 mol/L thiourea, 50 mmol/L Tris, and 25 mmol/L DTT). The

 

protein concentration of each sample was measured using the Bradford assay. The protein samples were stored at −80 °C for later use.

The urinary proteins were prepared using the filter-aided sample preparation (FASP) method (*Wisniewski et al., 2009*).Then, peptides were collected after enzymatic digestion using trypsin (Trypsin Gold, Promega, Fitchburg, WI, USA), desalted by Oasis HLB cartridges (Waters, Milford, MA, USA) and vacuum dried. The peptides were redissolved in 0.1% formic acid water and diluted to 0.5 µg/µL. A mixed peptide sample was prepared from each sample and separated using a high pH reversed-phase peptide separation kit (Thermo Fisher Scientific, Waltham, MA, USA); the resulting peptides were dried under, a vacuum and then redissolved in 0.1% formic acid in water for subsequent library construction. The iRT standard (Biognosys, Munich, Germany) was added to all identified samples for retention time normalization.

## LC–MS/MS analysis

The peptides were separated by using an EASY-nLC 1200 UPLC system (Thermo Fisher Scientific, Waltham, MA, USA). Peptides were dissolved in 0.1% formic acid in water, and 1 µg of each peptide sample was loaded on a PepMap column (75 µm × 2 cm, 3 µm, C18; Thermo Fisher). The eluate was loaded onto a reversed-phase analytical column (50 µm × 250 mm, 2 µm, C18; Thermo Fisher) with an elution gradient of 4–35% mobile phase B (80% acetonitrile + 0.1% formic acid + 20% water at a flow rate of 300 nL/min) for 90 min. For fully automated and sensitive signal processing, a calibration kit (iRT kit, Biognosys, Switzerland) was used with all samples at a concentration of 1:20 v/v. Then, fractions were analyzed by data-dependent acquisition (DDA)-MS with the following settings: 2.4 kV spray voltage, 60,000 Orbitrap primary resolution, 350–1,550 m/z scan range, 200–2,000 m/z secondary scan range, 30,000 resolution, 2 Da screening window, 30% HCD collision energy. The AGC target was 5e4, and the maximum injection time was 30 ms. The raw files were used to build a dataset and were analyzed by PD software (Proteome Discoverer 2.1; Thermo Fisher Scientific, Inc.).

The PD results were used to establish the data-independent acquisition (DIA) mode, and the m/z distribution density was used to calculate the window width and number. A single peptide sample was subjected to DIA mode, and each sample was repeated twice. Thirty samples were analyzed by DIA-MS. The liquid phase parameters were acquired in the same way as for the DDA database. The MS parameters of the mass spectrometry were set as follows: first level full scan at 60,000 resolution with a 350–1,550 m/z scan range, followed by a second level scan at 30,000 resolution with 39 screening windows, 30% HCD collision energy, AGC target of 1e6, and 50 ms maximum injection time. For the window calculation: the DDA results from the library acquisition were sorted into 39 groups based on m/z. The m/z range of each group is the window width of the collected DIA data. During the sample analysis, a mixture from each sample was analyzed after every seven samples for quality control (QC).

## Data analysis

MS data were processed and analyzed using Spectronaut software. The raw files acquired by DIA for each sample were imported for the library search. The false discovery rate (FDR)

of the proteins was less than 1%. A highly credible protein identification was indicated by $q < 0.01$, and all fragment ion peak areas of secondary peptides were used for protein quantification. The k-nearest neighbor (K-NN) method was used to fill missing values of protein abundance. Comparisons between two groups were performed by Student's t test. The differentially expressed proteins at Day 7 and 30 were screened by the following criteria: fold change $\geq 1.5$ or fold change $\leq 0.67$. Group differences at $p < 0.05$ were identified as statistically significant. Considering that omics datasets are large but the sample size is limited, the differences between two groups may be random (*Meng & Gao, 2020*). To confirm whether the differentially expressed proteins were indeed due to tumor resection, we randomly allocated the proteomic data for 10 samples at each time point. To determine that the differential proteins identified were due to random allocation, the same criteria for screening differential proteins were applied: fold change $\geq 1.5$ or fold change $\leq 0.67$ and $p < 0.05$. We used the Wu Kong platform (https://www.omicsolution.com/wkomics/main/) relative orthogonal signal-corrected partial least squares discriminant analysis (OPLS-DA). The differentially expressed proteins at different time points were functionally annotated using DAVID (https://david.ncifcrf.gov/), and $p < 0.05$ was considered to indicate statistical significance. Venn diagrams indicating the overlap of differentially expressed proteins were plotted using software PyCharm 2018.2 (Edu).

# RESULTS

## Establishment of animal models

All three groups underwent resection 7 or 8 days after tumor cell inoculation, when the tumors were palpable, and urine was collected on Days 7 and 30 after resection. The procedure is shown in Fig. 2. The animal model was successfully established, and tumor size on Day 7 or 8 is shown in Table 1. No tumor recurrence was seen 90 days after resection in the tumor-resected group; Therefore, the resection was considered successful. No abnormalities were seen in the healthy control group.

## Analysis of urine proteome changes

Analysis of the results of the urine proteome at two time points for the three groups led to the identification of a total of 405 proteins in all samples. The MS proteomics data are available at iProX project PXD034552.

OPLS-DA showed a clear distinction between the tumor-resected group and tumor-bearing group at 7 and 30 days after surgery. The R2 (overall goodness of fit) and Q2 (overall goodness of prediction) values were calculated, and the results are shown in Fig. 3. On Day 7 after surgery, the R X, R Y and Q Y values were 0.369, 0.993 and 0.788, respectively, indicating good model fit and accuracy. The variable importance in projection (VIP) score was calculated for the differentially expressed proteins, and all 20 differential expressed proteins met the criteria of VIP $> 1.0$. Similarly, on Day 30 after surgery, the R X, R Y and Q Y values were 0.435, 0.968 and 0.766, respectively, indicating good model fit and accuracy. The VIP scores for all 33 the differentially expressed proteins met the criteria of VIP $> 1.0$. This model demonstrates the possibility of distinguishing whether MC38 tumors have been resected.

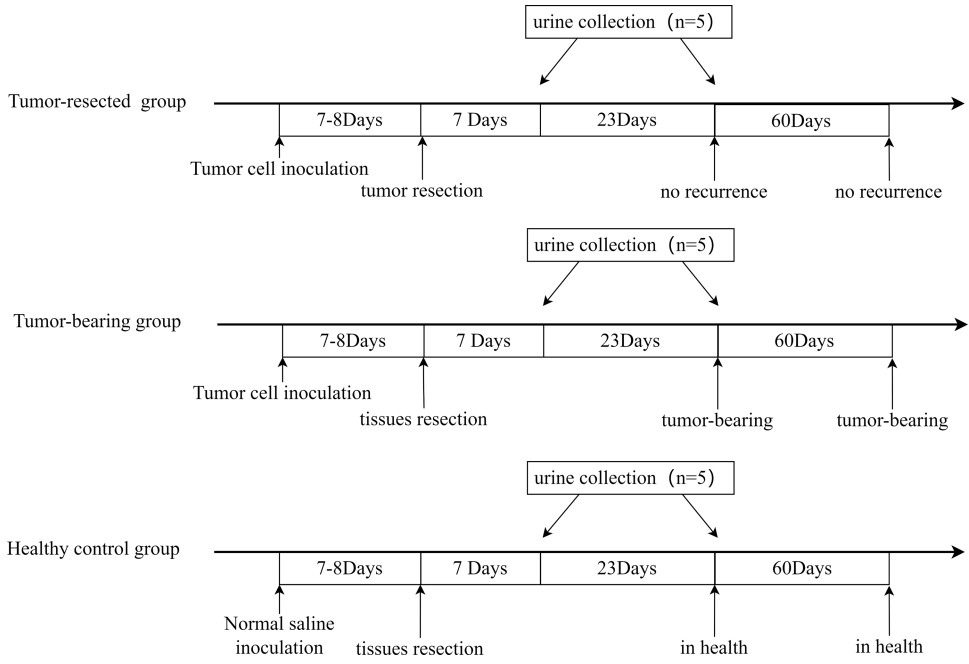

**Figure 2  Animal model establishment process and the time point of urine collection.**

**Table 1  Tumor size (expressed as the mean diameter).**

| | | | Tumor-resected group ($n = 5$) | | |
|---|---|---|---|---|---|
| No. | 1 | 2 | 3 | 4 | 5 |
| Size (mm) | 4.2 | 5.0 | 5.2 | 4.6 | 4.5 |
| | | | Tumor-bearing group ($n = 5$) | | |
| No. | 6 | 7 | 8 | 9 | 10 |
| Size (mm) | 4.6 | 3.7 | 6.0 | 4.8 | 3.4 |
| | | | Healthy control group ($n = 5$) | | |
| No. | 11 | 12 | 13 | 14 | 15 |
| Size (mm) | – | – | – | – | – |

In total, the levels of 20 proteins (Table 2) were significantly different between the tumor-resected group and the tumor-bearing group at Day 7 after resection, of which nine were upregulated and 11 were downregulated. There were 125 random allocations, and the average number of differentially expressed proteins in all random combinations was 3.44. indicating that the false-positive rates were 17.2%. Six biological processes were enriched by the differentially expressed proteins on Day 7 (Fig. 4), including circadian rhythm, Notch signaling pathway, leukocyte cell–cell adhesion, and heterophilic cell–cell adhesion *via* plasma membrane cell adhesion molecules. In the cellular component category (Fig. 4B), all the differentially expressed proteins were enriched in the secreted and lysosome terms. In the molecular function category (Fig. 4C), serine protease and hydrolase were overrepresented.

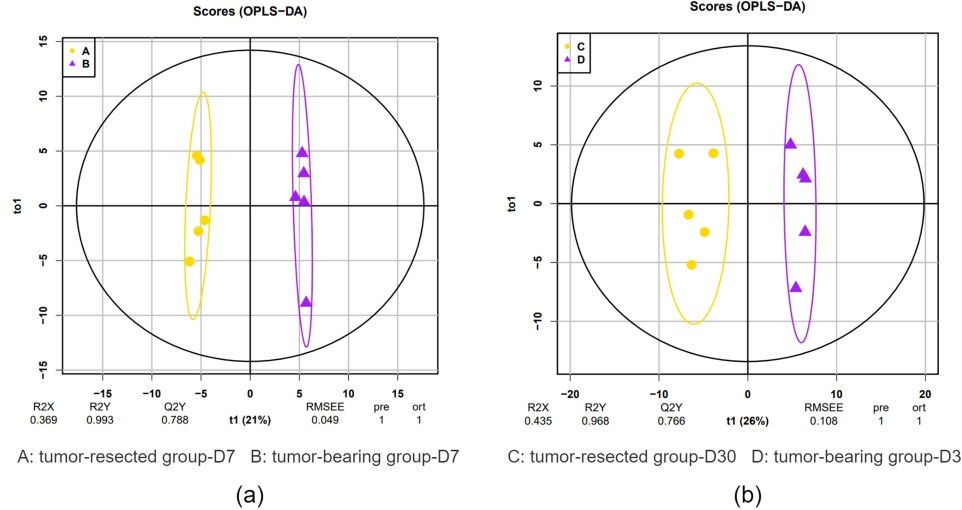

A: tumor-resected group-D7   B: tumor-bearing group-D7     C: tumor-resected group-D30   D: tumor-bearing group-D30

(a)                          (b)

**Figure 3 OPLS-DA of the urine samples of resection and no resection mice.** (A) OPLS-DA of all urine proteins from tumor-resected and tumor-bearing group on Day 7; (B) OPLS-DA of all urine proteins from tumor-resected and tumor-bearing group on Day 30.

**Table 2 Differentially expressed proteins between tumor-resected group and tumor-bearing group on Day 7.**

| Protein accessions | Protein descriptions | Human ortholog | *p* value | Fold change |
|---|---|---|---|---|
| Q62266 | Cornifin-A | – | 4.21E−02 | 2.883 |
| P00687 | Alpha-amylase 1 | P04746 | 3.52E−02 | 2.130 |
| Q91X17 | Uromodulin | P07911 | 2.80E−03 | 2.092 |
| O35657 | Sialidase-1 | Q99519 | 2.50E−04 | 2.002 |
| P0CG49 | Polyubiquitin-B | P0CG47 | 1.92E−02 | 1.968 |
| Q80X71 | Transmembrane protein 106B | Q9NUM4 | 6.97E−05 | 1.949 |
| Q60590 | Alpha-1-acid glycoprotein 1 | P02763 | 1.30E−02 | 1.825 |
| P81117 | Nucleobindin-2 | P80303 | 1.14E−02 | 1.727 |
| P18761 | Carbonic anhydrase 6 | P23280 | 4.58E−02 | 1.541 |
| Q07797 | Galectin-3-binding protein | Q08380 | 3.74E−02 | 0.669 |
| P06869 | Urokinase-type plasminogen activator | P00749 | 1.57E−02 | 0.668 |
| Q05793 | Basement membrane-specific heparan sulfate proteoglycan core protein | P98160 | 4.62E−02 | 0.666 |
| P29533 | Vascular cell adhesion protein 1 | P19320 | 2.20E−02 | 0.662 |
| P05533 | Lymphocyte antigen 6A-2/6E-1 | – | 3.99E−02 | 0.661 |
| P0CW02 | Lymphocyte antigen 6C1 | – | 9.07E−04 | 0.660 |
| Q9ESD1 | Prostasin | Q16651 | 9.78E−03 | 0.659 |
| P15947 | Kallikrein-1 | P06870 | 1.95E−03 | 0.632 |
| Q9Z0J0 | NPC intracellular cholesterol transporter 2 | P61916 | 1.93E−03 | 0.578 |
| O09051 | Guanylate cyclase activator 2B | Q16661 | 5.10E−03 | 0.532 |
| Q9EP95 | Resistin-like alpha | Q9BQ08 | 5.79E−03 | 0.421 |

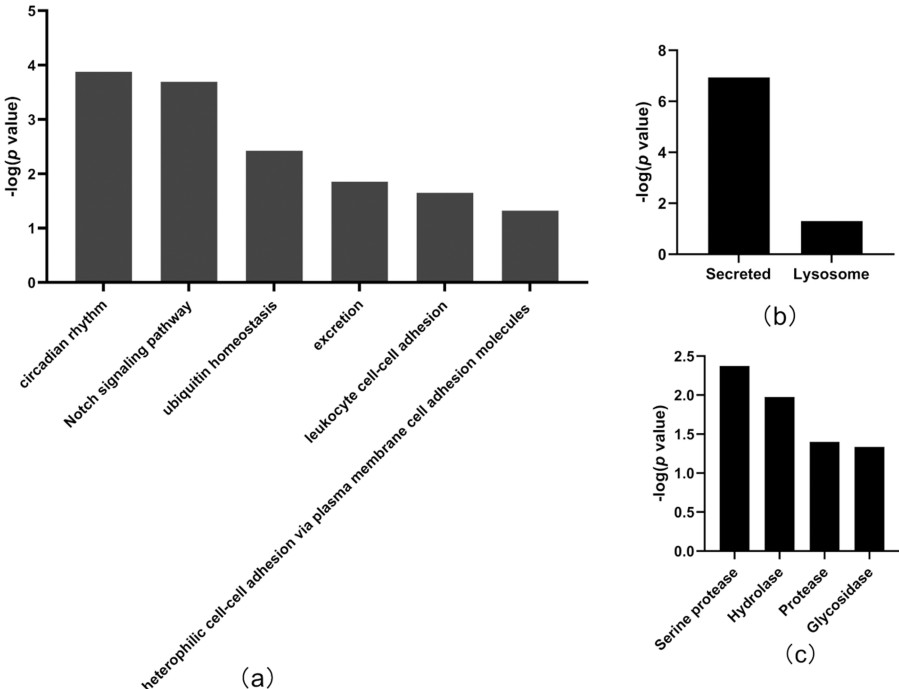

**Figure 4** **Functional annotation of differentially expressed proteins between tumor-resected group and tumor-bearing group on Day 7.** *X*-axis represents biological process (A), cellular component (B) and molecular function (C) in Day 7 between tumor-resected group and tumor-bearing group. *Y*-axis represents the *P* values (−log 10) in the annotation categories. In this study, *p*-value is equal or smaller than 0.05 should be considered strongly enriched in the annotation categories.

Thirty-three differentially expressed proteins were identified between the tumor-resected group and the tumor-bearing group at Day 30 after surgery (Table 3), of which 13 were upregulated and 20 were downregulated, enriched in 19 biological processes (Fig. 5). There were 125 random allocations, and the average number of differential proteins in all random combinations was 3.76. indicating that the false-positive rates were 11.4%. Based on the data for Day 30, the biological processes on Day 30 included cell adhesion, regulation of protein localization to the cell surface, complement activation, alternative pathway, immune system processes, positive regulation of glucose metabolic processes, mitochondrion morphogenesis, and angiogenesis. In the cellular component category (Fig. 5B), the majority of the differentially expressed proteins were secreted. In the molecular function category (Fig. 5C), growth factor binding was overrepresented.

Compared to the healthy control group, there were 22 and 8 differentially expressed proteins (Tables 4 and 5) in the tumor-resected group on Days 7 and 30 after surgery. Eight differentially expressed proteins on Day 30 were not sufficient for enrichment analysis of biological processes. There were 125 random allocations, and the average number of differentially expressed proteins in all random combinations was 3.77, indicating a false-positive rate of 47%. These data suggest that the difference in the physiological

**Table 3  Differentially expressed proteins between tumor-resected group and tumor-bearing group on Day 30.**

| Protein accessions | Protein descriptions | Human ortholog | p value | Fold change |
|---|---|---|---|---|
| Q8VED5 | Keratin, type II cytoskeletal 79 | Q5XKE5 | 2.81E−02 | 3.026 |
| P00687 | Alpha-amylase 1 | P04746 | 2.16E−02 | 2.778 |
| Q922U2 | Keratin, type II cytoskeletal 5 | P13647 | 1.10E−02 | 2.732 |
| Q61781 | Keratin, type I cytoskeletal 14 | P02533 | 2.12E−02 | 2.685 |
| P11591 | Major urinary protein 5 | – | 2.94E−02 | 2.292 |
| Q6NXH9 | Keratin, type II cytoskeletal 73 | Q86Y46 | 4.80E−02 | 1.997 |
| P03953 | Complement Factor D | P00746 | 8.82E−04 | 1.784 |
| O55186 | CD59A glycoprotein | P13987 | 2.11E−02 | 1.671 |
| Q61581 | Insulin-like growth factor-binding protein 7 | Q16270 | 2.78E−03 | 1.598 |
| P09803 | Cadherin-1 | P12830 | 5.21E−03 | 1.542 |
| Q9JJS0 | Signal peptide, CUB and EGF-like domain-containing protein 2 | Q9NQ36 | 5.10E−03 | 1.535 |
| P11589 | Major urinary protein 2 | – | 3.36E−02 | 1.523 |
| P01132 | Pro-epidermal growth factor | P01133 | 3.61E−03 | 1.516 |
| Q07456 | Protein AMBP | P02760 | 2.47E−03 | 0.652 |
| P47878 | Insulin-like growth factor-binding protein 3 | P17936 | 8.47E−03 | 0.628 |
| P04186 | Complement factor B | P00751 | 4.87E−02 | 0.627 |
| Q9WTR5 | Cadherin-13 | P55290 | 3.56E−02 | 0.617 |
| P11276 | Fibronectin | P02751 | 2.30E−03 | 0.594 |
| Q61129 | Complement factor I | P05156 | 1.51E−02 | 0.579 |
| Q921W8 | Secreted and transmembrane protein 1A | Q8WVN6 | 3.60E−04 | 0.575 |
| O88968 | Transcobalamin-2 | P20062 | 2.67E−02 | 0.538 |
| Q91X72 | Hemopexin | P02790 | 1.94E−03 | 0.473 |
| P0CW02 | Lymphocyte antigen 6C1 | – | 1.21E−03 | 0.466 |
| Q8K4G1 | Latent-transforming growth factor beta-binding protein 4 | Q8N2S1 | 1.15E−02 | 0.454 |
| O09051 | Guanylate cyclase activator 2B | Q16661 | 2.69E−03 | 0.442 |
| P25119 | Tumor necrosis factor receptor superfamily member 1B | P20333 | 2.37E−02 | 0.431 |
| Q9EP95 | Resistin-like alpha | Q9BQ08 | 9.20E−03 | 0.389 |
| Q05793 | Basement membrane-specific heparan sulfate proteoglycan core protein | P98160 | 4.91E−04 | 0.378 |
| P29533 | Vascular cell adhesion protein 1 | P19320 | 1.16E−02 | 0.352 |
| O08997 | Copper transport protein ATOX1 | O00244 | 1.02E−02 | 0.318 |
| Q91VW3 | SH3 domain-binding glutamic acid-rich-like protein 3 | Q9H299 | 3.02E−02 | 0.297 |
| Q4KML4 | Costars family protein ABRACL | Q9P1F3 | 3.05E−02 | 0.254 |
| Q61646 | Haptoglobin | P00739 | 2.19E−02 | 0.214 |

condition of mice in the tumor-resected group compared to that of mice in the healthy control group was smaller after 30 days of recovery.

The overlap of these differentially expressed proteins is shown in the Venn diagram in Fig. 6. The six proteins were significantly different between the tumor-resected group and the tumor-bearing group at both Day 7 and Day 30, including vascular cell adhesion protein 1, basement membrane-specific heparan sulfate proteoglycan core protein, resistin-like

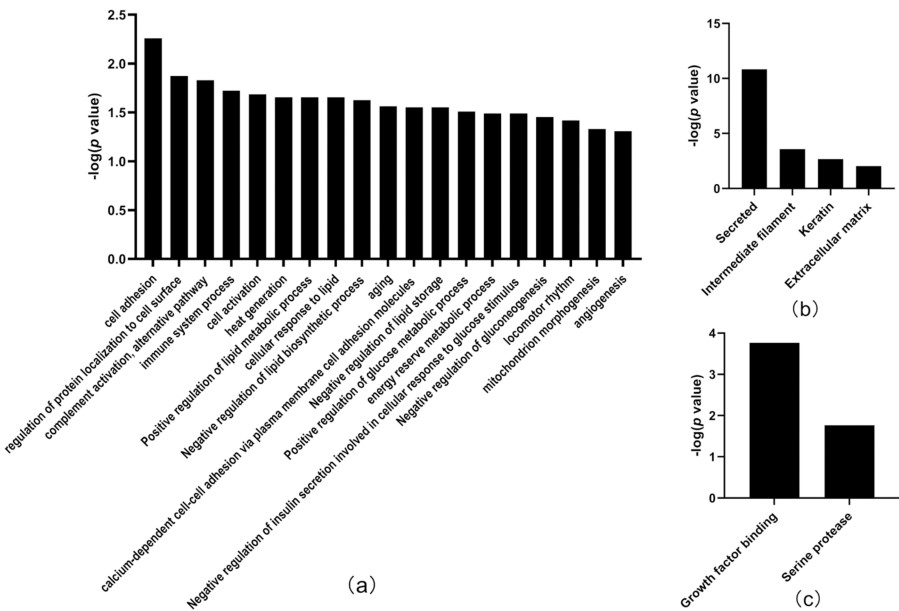

**Figure 5** **Functional annotation of differentially expressed proteins between tumor-resected group and tumor-bearing group on Day 30.** *X*-axis represents biological process (A), cellular component (B) and molecular function (C) in Day 30 between tumor-resected group and tumor-bearing group. *Y*-axis represents the *P* values (−log 10) in the annotation categories. In this study, *p*-value is equal or smaller than 0.05 should be considered strongly enriched in the annotation categories.

alpha, guanylate cyclase activator 2B, lymphocyte antigen 6C1, and alpha-amylase 1. The changes of these six proteins were consistent at two time points. The vascular cell adhesion protein 1, basement membrane-specific heparan sulfate proteoglycan core protein, resistin-like alpha, guanylate cyclase activator 2B, and lymphocyte antigen 6C1 protein in the urine of the tumor-resected group decreased, while the alpha-amylase 1 protein increased. Five proteins were at identical levels in the tumor-resected group and tumor-bearing group and in the tumor-resected group and health control group at Day 30, including CD59A glycoprotein, lymphocyte antigen 6C1, fibronectin, basement membrane-specific heparan sulfate proteoglycan core protein, keratin, type II cytoskeletal 5.

## DISCUSSION

In this study, an MC38 tumor-bearing mouse model was established. Compared to tumor-bearing mice, those that underwent tumor resection had a total of 20 and 33 differentially expressed proteins on Day 7 and 30 after resection respectively. Randomized grouping statistical analysis showed that more than 83% of the differentially expressed proteins identified in this study were caused by tumor resection rather than by random allocation. The difference in the urine proteomes of the control group and the tumor-resected group was smaller 30 days after resection. The OPLS-DA results indicated that urine proteins have a strong ability to discriminate the tumor-resected group and the tumor-bearing group at Days 7 and 30.

**Table 4  Differentially expressed proteins between tumor-resected group and the healthy control group on Day 7.**

| Protein accessions | Protein descriptions | Human ortholog | $p$ value | Fold change |
|---|---|---|---|---|
| Q9EP95 | Resistin-like alpha | Q9BQ08 | 2.21E−02 | 2.413 |
| Q01279 | Epidermal growth factor receptor | P00533 | 8.93E−03 | 1.876 |
| Q9ESD1 | Prostasin | Q16651 | 3.15E−03 | 1.836 |
| P00688 | Pancreatic alpha-amylase | P04746 | 1.47E−02 | 1.765 |
| C0HKG5 | Ribonuclease T2-A | – | 5.51E−03 | 1.728 |
| P15947 | Kallikrein-1 | P06870 | 1.12E−03 | 1.713 |
| Q9Z0J0 | NPC intracellular cholesterol transporter 2 | P61916 | 2.99E−02 | 1.540 |
| Q8K1H9 | Odorant-binding protein 2a | Q9NY56 | 4.72E−02 | 0.642 |
| P18761 | Carbonic anhydrase 6 | P23280 | 3.08E−02 | 0.631 |
| P11589 | Major urinary protein 2 | – | 3.04E−02 | 0.631 |
| P09470 | Angiotensin-converting enzyme | P12821 | 4.90E−02 | 0.606 |
| P97426 | Eosinophil cationic protein 1 | P10153 | 3.18E−02 | 0.603 |
| P07361 | Alpha-1-acid glycoprotein 2 | P02763 | 7.98E−03 | 0.589 |
| P60710 | Actin | P60709 | 3.24E−02 | 0.586 |
| Q5FW60 | Major urinary protein 20 | – | 2.17E−02 | 0.575 |
| P81117 | Nucleobindin-2 | P80303 | 9.76E−03 | 0.574 |
| Q08423 | Trefoil factor 1 | P04155 | 1.36E−02 | 0.533 |
| Q91X17 | Uromodulin | P07911 | 1.35E−03 | 0.500 |
| Q60590 | Alpha-1-acid glycoprotein 1 | P02763 | 4.10E−03 | 0.479 |
| P0CG49 | Polyubiquitin-B | P0CG47 | 1.21E−02 | 0.462 |
| Q62266 | Cornifin-A | – | 2.97E−02 | 0.289 |
| Q922U2 | Keratin, type II cytoskeletal 5 | P13647 | 3.58E−02 | 0.269 |

**Table 5  Differentially expressed proteins between tumor-resected group and healthy control group on Day 30.**

| Protein accessions | Protein descriptions | Human ortholog | $p$ value | Fold change |
|---|---|---|---|---|
| Q922U2 | Keratin, type II cytoskeletal 5 | P13647 | 4.10E−02 | 2.102 |
| P60710 | Actin, cytoplasmic 1 | P60709 | 1.63E−02 | 1.695 |
| O55186 | CD59A glycoprotein | P13987 | 4.71E−02 | 1.502 |
| P11276 | Fibronectin | P02751 | 1.79E−02 | 0.617 |
| P0CW02 | Lymphocyte antigen 6C1 | – | 2.46E−02 | 0.609 |
| P06869 | Urokinase-type plasminogen activator | P00749 | 2.43E−02 | 0.576 |
| Q05793 | Basement membrane-specific heparan sulfate proteoglycan core protein | P98160 | 1.04E−02 | 0.526 |
| Q91WR8 | Glutathione peroxidase 6 | P59796 | 8.99E−03 | 0.480 |

Related to the biological process analysis results on Day 7, some studies have shown that circadian rhythms are relevant to both wound healing and tumor growth; specifically, wound healing is accompanied by inflammation, leukocyte transport, and tissue remodeling, and that some of these proteins are involved in a circadian-driven chronological coordination mechanism (*Cable, Onishi & Prendergast, 2017*; *Sherratt et al., 2019*). The Notch signaling pathway is involved in regulating the behavior of

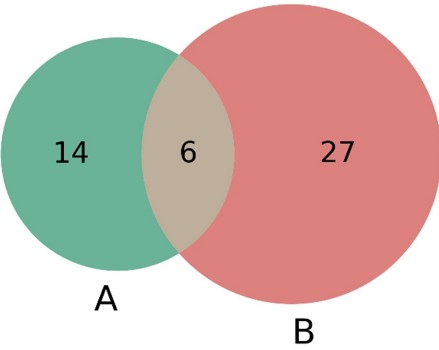

A: tumor-bearing D7 vs tumor-resected D7    B: tumor-bearing D30 vs tumor-resected D30

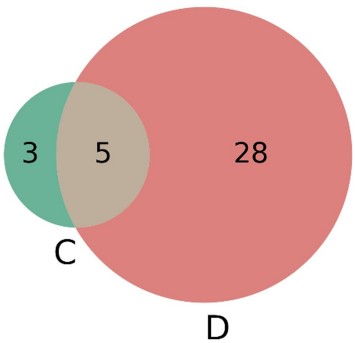

C: tumor-resected D30 vs healthy control D30    D: tumor-bearing D30 vs tumor-resected D30

**Figure 6** **(A–D) Overlap evaluation of differentially expressed proteins between the groups/time-points.**

multidimensional subcutaneous tumor (*Hu et al., 2009*). Notch signaling is associated with oncogene expression, is potentially a target of tumor suppressors and can affect tumor cell proliferation, differentiation, apoptosis and genomic instability (*Bernardo et al., 2021*). In lymphatic endothelial cells, regulation of the cell expression of the integrin ligands ICAM-1 and VCAM, which control leukocyte transition, induces enhanced leukocyte adhesion to generate an immune response against tumors (*Rodriguez-Ruiz et al., 2017*). A previous study showed that tumor expression of VCAM1 represents a novel mechanism of immune evasion (*Lin et al., 2007*). Our results showed that VCAM1 was downregulated in the tumor-resected group compared to the tumor-bearing group, which may imply that the resection eliminated VCAM-1-expressing tumor cells. Guanylate cyclase activator 2B, which is involved in the excretion pathway, and it has been reported to predict and assess the survival of colorectal cancer patients (*Pan et al., 2017*). Furthermore, some differentially expressed proteins are not involved in the enriched pathways, but are equally important. Suppression of NUCB-2 inhibited tumor nodule formation in a mouse colon cancer model (*Kan et al., 2016*). The ability of galectin 3 to support cancer cell survival by intracellular and extracellular mechanisms suggests that this molecule is an important component of

the tumor microenvironment (*Ruvolo, 2016*; *Cho et al., 2021*). Cornifin-A is expressed in mouse skin and Cornifin-A protein is upregulated in the differentiated area of squamous cell carcinoma (*Owens et al., 1996*). *In vitro* studies have shown that resistin-like alpha has functions in angiogenesis and tissue remodeling and is required in the wound healing process following surgery-related tissue injury (*Pine, Batugedara & Nair, 2018*).

Among the biological processes identified at Day 30, cell adhesion and regulation of protein localization to the cell surface are relevant to tumor growth; cadherin and integrin are particularly important, as they act as receptors for ligand activation and trigger relevant signaling through changes in the physical environment (*Janiszewska, Primi & Izard, 2020*). Immune-related pathways are frequently altered during tumor growth; for example, complement activation in the tumor microenvironment enhances tumor growth and migration (*Afshar-Kharghan, 2017*). Haptoglobin is involved in the immune response process. A study showed that haptoglobin-knockout mice have more tumor masses, suggesting that acute-phase reactants inhibit tumorigenesis, possibly by suppressing inflammatory responses (*Barbour et al., 2001*). There is evidence that changes in calcium-dependent cell adhesion molecules, such as cadherin-1 (CDH1), are associated with tumors. Cadherin downregulation can be used in the diagnosis and prognosis of epithelial cancer (*Van Roy, 2014*). Cadherin-13 (CDH13) plays a regulatory role in tumor growth and promotes angiogenesis (*Andreeva & Kutuzov, 2010*; *Van Roy, 2014*). In addition, tumor cells secrete high levels of angiogenic factors, which contribute to the generation of abnormal vascular networks; thus, the inhibition of angiogenesis a key strategy for cancer therapy (*Viallard & Larrivee, 2017*). In addition, the 33 differentially expressed proteins offer some interesting clues. Previous findings suggest a link between the copper transport protein ATOX1 and NADPH oxidase in inflammatory neovascularization, indicating that ATOX1 is a potential therapeutic target for the treatment of ischemic disease (*Chen et al., 2015*). Another study found that TNF is not just an effector but also an initiator of inflammatory Th cell differentiation. Tumor necrosis factor receptor superfamily member 1B may mediate the generation of inflammatory Th1 cells (*Alam et al., 2021*). Hemopexin may block heme-driven tumor growth and metastasis (*Canesin et al., 2020*). The endothelial cell surface CUB and EGF-like domain-containing protein 2 (SCUBE2) is a coreceptor for vascular endothelial growth factor receptor 2 (VEGFR2) that is required for pathological tumor angiogenesis (*Lin et al., 2018*).

Compared to the healthy control group, the tumor-resected group had eight differentially expressed proteins changes on Day30. This result does not mean that the physical condition of mice in the tumor-resected group returned to normal at 30 days after surgery, but rather that the difference from mice in the healthy control group narrowed. Overall, the urinary proteome can indicate whether an MC38 subcutaneous tumor has been completely resected.

This preliminary study included a limited number of mice, and the differentially expressed proteins identified in this study require further verification in a large number of human urine samples. Additional experiments can be designed, for example, to add a partial tumor-resected group. The urine proteomes after tumor resection were different, suggesting that urine proteomics may distinguish whether a tumor is completely removed. Additionally, this study represents a starting point for future studies of the urinary proteome

after tumor resection. This study provides a new research direction for surgeons to assess tumor resection.

## CONCLUSIONS

Our results revealed that urine proteomics can distinguish between MC38 tumor-bearing mice and tumor-resected mice. These findings may provide a new strategy for clinical studies.

## ACKNOWLEDGEMENTS

I would like to acknowledge Professor. Wang Sheng and his student Weijian Zhao from the Faculty of Environment and Life, Beijing University of Technology for providing MC38 cell line.

### Funding

The authors received funding from the National Key Research and Development Program of China (2018YFC0910202), the Fundamental Research Funds for the Central Universities (2020KJZX002), and Beijing Normal University (11100704). The funders had no role in study design, data collection and analysis, decision to publish, or preparation of the manuscript.

### Grant Disclosures

The following grant information was disclosed by the authors:
National Key Research and Development Program of China: 2018YFC0910202.
Fundamental Research Funds for the Central Universities: 2020KJZX002.
Beijing Normal University: 11100704.

### Competing Interests

The authors declare there are no competing interests.

### Author Contributions

- Ziqi Heng conceived and designed the experiments, performed the experiments, analyzed the data, prepared figures and/or tables, authored or reviewed drafts of the article, and approved the final draft.
- Chenyang Zhao performed the experiments, prepared figures and/or tables, and approved the final draft.
- Youhe Gao conceived and designed the experiments, authored or reviewed drafts of the article, and approved the final draft.

### Animal Ethics

The following information was supplied relating to ethical approvals (*i.e.*, approving body and any reference numbers):

The experiment was approved by the ethics and animal welfare committee of Beijing Normal University (Approval ID: CLS-EAW-2020-0344).

## Data Availability

Data is available at iProX: PXD034552.

## Supplemental Information

Supplemental information for this article can be found online at http://dx.doi.org/10.7717/peerj.14737#supplemental-information.

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
