# Peer review of "Comparison of urine proteomes from tumor-bearing mice with those from tumor-resected mice"

_PeerJ, doi:10.7717/peerj.14737_

## Round 0.1 · original submission · Major Revisions

Please address the concerns of all reviewers and revise the manuscript accordingly.

Reviewer 1 ·

Basic reporting

a. The writing is unclear in a number of areas (e.g. line 63, line 96, line 170, line 182: R^X vs R2X in Fig. 3, line 195, line 247). The methods section in particular was problematic (line 124, line 129-130: column cat ID?, line 137: one of multiple unexplained abbr., line 154, line 158-160).
b. The introduction did not provide a fair assessment of the background literature. The authors exclusively referenced their own work.
c. Insufficient references in results.
d. In contrast to the introduction, the authors ignore their own work in the discussion, and also did not attempt to compare proteomes to those reported in other studies.
e. The raw data was uploaded to iProX but the .RAW files does not appear to be annotated. I was unable to find the processed data displayed in the figures/tables.
f. In methods, results and Fig. 2. No resection was performed on controls, so ‘mock surgery’ may be a more appropriate label. It is unclear exactly what defines ‘recurrence’ / ‘no recurrence’, is it simply whether a tumor was observed?
g. Fig. 4 and 5. P-values presents statistical significance but not biological significance, and maybe not the best way to display these data.
h. What are the number of proteins per sample? (line 178)
i. Scientifically vague (“basically recovered”, line 233)
j. Line 162: the website for the ‘Wu Kong’ platform is not in English
k. Line 164: the reference provided was for StatsPro and does not appear to support OPLS-DA? I was attempting to use the web version.

Experimental design

a. The article appears to be one of multiply work by the authors related to urinary proteomics and cancer using similar scientific methods. The specific tumor model differentiate this manuscript from their other works.
b. The authors designed an experiment with three groups: 1) healthy, 2) tumor-bearing, and 3) tumor-resected. This would presumably allow them to 1) assess whether a subcutaneous tumor model using MC38 cells would causes observable changes in urine proteome, and 2) whether complete resection of the tumor would prevent such changes. They sampled urine twice, at 7 and 30 days after resection or mock surgery. Urine proteome was analyzed using LC-MS/MS.
c. However, rather than exploring all comparisons, only 3 comparisons were presented: 1) tumor vs resected at day 7 (text, Fig. 4, Table 2), 2) tumor vs resected at day 30 (text, Fig. 5, Table 3), and 3) healthy vs resected (Table 4, mentioned only in abstract).

Validity of the findings

a. The section describing data analysis was particularly unclear and there may be issues with the statistical analysis. It is unclear why pairwise comparisons were performed rather than using tests designed for multiple group comparisons (line 157). Performing ANOVA for two groups is equivalent to a t-test. It was unclear whether p-values were corrected for multiple comparisons.
b. It was unclear why specifically fold changes of 1.5 and 0.67 were used as the thresholds. A statistical power calculation would have helped with assessing whether the small sample size (n=5 per group) was appropriate for detecting changes at the stated fold change.
c. It is unclear whether random allocation is an appropriate method to estimate false discovery rate. False discovery rate / Type I error is estimated by the p-value. If the p-values were not corrected for multiply comparison, then about 5% of the all unchanged proteins would be falsely detected as different (if the number of compared proteins is 405 then: 405*0.05 ≈ 20). Regardless, it is statistically incorrect to state with certainly “at least XX% of the differential proteins…” (line 199, line 211, line 230).
d. It is somewhat concerning that not all comparisons were reported. E.g. comparison between healthy animals between 7 and 30 days would provide an estimate of false discovery rate as there should be no differences between the groups. Comparison between healthy and tumor-bearing animals at both time points would provide ‘cleaner’ comparison without the added complication of the resection. Comparison of the healthy and tumor-resected animals would be interesting to investigate whether there were persistent effect of carrying the tumor for a short time.
e. Related to the last point. Generating a Venn diagram of the differentially-expressed proteins between all the groups/timepoints would be critical to fully explore the data. For example 7 of the 8 differentially-expressed proteins between the healthy and resection group (Table 4) was also observed to change in the two other comparisons (Table 2 and 3).
f. In the abstract, results, point 3 states no significant difference but Table 4 shows 8 differentially-expressed proteins between healthy and resection group at day 30.
g. In some areas, the authors strongly suggest to readers that this study provided evidence that urine proteome can be used to detect incomplete tumor resection (line 18, line 37, line 259, 267). The authors then partially explained in the final discussion (line 263) that this is not quite true and further experiments are required. The authors should probably be more conservative and keep in line with what is supported by the experimental data.
h. The title clearly stated comparison of tumor-bearing vs tumor-resected mice. However, in the text those two groups are named nonresection and complete resection, which I believe is less clear and suggests there is a partial resection group.

Reviewer 2 ·

Basic reporting

This manuscript used mouse tumor models to study the changes of urine proteome associated with tumor resection. Compared with tumor-bearing mice, changes of protein expression as well as certain cellular processes were observed in tumor-resected mice. The authors stated this could be a new strategy for clinical studies.

However, this manuscript does not provide enough background regarding why urine serves as a good source for biomarkers than blood. Only one reference was cited to support this statement. The authors should also explain the rationale of why a mouse model of colorectal cancer was chosen, instead of other cancer models, for this study. The English writing needs to be improved in order to provide enough background as well as to understand the proteomic results. The figures 3, 4, and 5 need to be revisited by adding figure legend details, labeling of panels, what the x- and y-axis stand for, and why only certain numbers of proteins or cellular processes were shown. I suggest major revision and my comments are listed below.

Experimental design

1. Please explain why the authors decided to use tumor cell inoculation to generate the mouse model for this study. Is there any concern that this subcutaneous model may not reflect the complexity and functions of cells in original colorectal cancer and the corresponding environment? Could genetically engineered mouse models be a better model system?
2. How is the complete resection group is defined?
3. Please explain why urine samples were collected at day 7 and day 30. For the study mentioned in the introduction section (line 50-61), the urine proteome from walker 256 tumor-bearing rats was carried out at different time points. What factors contribute to this time point selection?
4. Please add details in the “data analysis” method section regarding how data were analyzed using the DAVID database. How to draw the conclusion about what enriched biological processes are and how many processes are involved? In line 200, the authors stated 6 processes were enriched on day 7 sample. Does it mean only 6 were enriched from data analysis or any further selection were performed. Same question for line 212-215.

Validity of the findings

1. I suggest the authors rewrite the paragraph line 218-225. Why are the differential proteins identified on Day 30 sample not sufficient to enrich biological processes (line 220)? Why does a false-positive rate of 47% for differential proteins mean the physiological condition of the complete resection group is similar to the healthy control group?
2. In the conclusion section line 259, the authors concluded that urine proteome can reflect whether the tumor has been resected. I think more data will be required to support this generalization. As this study only used colorectal cancer model, the data shown in this manuscript showed weak connection between urine proteome and all kinds of tumor.

Additional comments

In general, the introduction section needs more backgrounds or related literature reviews about what stable internal environment (line 44) means and why urine is a suitable source for biomarkers. In addition to my comments listed above, can the authors comment on the changes of urine proteome comparing day 7 and day 30 samples? Do the differential proteins identified on day 7 become incomparable between groups on day 30? If the proteins identified from different time points differ, how can this result help clinical studies? Specifically, how can this study help narrow down to certain biomarkers to use in a specified time points? Furthermore, the stages of cancer is not discussed in this manuscript. Whether original cancer or metastatic cancer limit the application of urine proteome for understanding the results of tumor resection could be worth discussed.

Reviewer 3 ·

Basic reporting

I suggest the authors contact a professional editing service to improve the English language, for example, the text of lines 25-26, 46-48, could be improved to make it clear and accurate. In another example in line 266, the authors claim that “This study provides a new method for surgeons to evaluate tumor resection.” I find this is a bit of a stretch to jump from the mouse model to clinical applications.
Also, the Introduction section could be improved. For example, a reference is needed for your claim in lines 45-47.

Experimental design

I believe the primary research reported in this paper is within the scope of the journal. The questions were well defined and meaningful, and the findings have potential applications in clinical studies.

Validity of the findings

no comment

---

## Round 0.2 · Major Revisions

As you can see, the reviewers are not satisfied by your responses and revision. Therefore, I am returning the manuscript for another round of major revision. Please address the remaining concerns of the reviewers and amend your manuscript accordingly.

Reviewer 1 ·

Basic reporting

a) There are still English issues after use of a language editing service
i. Issues with clarity. E.g. Line 80 “… we generated additional evidence…” The rest of the paragraph exclusively cites the work of others, not evidence that the authors generated. Line 237. “The levels of six proteins…”
ii. Issues with grammar. E.g. Line 114 “…control group was received an injection…”
iii. Spelling mistakes. E.g. Line 138. “prptides”, Line 149 “acquisiton”, Line 208 ”defferent”, Line 309 “norma”, “micce”.
b) (previously point 1b) For the introduction, the authors claim there are only a few studies perform by other labs that are relevant to this study. I would suggest studies related to urine proteome and (colorectal) cancer are worth discussing to help put the author’s series of work into perspective. This would also address the first point raised by Reviewer 2. E.g. “Urinary Proteomics Profiles Are Useful for Detection of Cancer Biomarkers and Changes Induced by Therapeutic Procedures” by Ferrari et al. (10.3390/molecules24040794) or “Noninvasive urinary protein signatures associated with colorectal cancer diagnosis and metastasis” by Sun et al. (10.1038/s41467-022-30391-8).
c) (previously point 1d) For the discussion, I expected the authors to compare the results to their previous study as mentioned in the introduction. Wu, J., Z. Guo and Y. Gao (2017). "Dynamic changes of urine proteome in a Walker 256 tumor-bearing rat model." Cancer Med 6(11): 2713-2722 DOI: 10.1002/cam4.1225.
d) (previously point 1f) My previous comments for Fig 2 was in relation to the labels and not the group naming. For the middle figure, should it be ‘tissue resection’ rather than ‘tumor resection’? For the bottom figure, I’m not sure if ‘no recurrence’ is correct or clear as there were no tumor ‘occurrence’. Should the ‘no recurrence’ labels at 30 days post resection be removed?
e) (previously point 1g) For Fig 4. And 5. I would assume that the ‘fold change’ would be more related to biological significance than p-values.
f) (previously point 1j) I’m not sure if this is an issue per journal policy but https://www.omicsolution.org/wkomics/main/ is still not in English.
g) Table 1, 2, and 3. The group names need to be updated from “Complete resection group” to “resection group”.
h) I agree with Reviewer 2 that more labels should be added to Fig 3.
i) Line 196. Should that be “… a total of 405 unique proteins …”?
j) Table 4 and 5 have identical captions.

Experimental design

a) (previously point 2c, 3a) The authors do not believe they have performed multiple comparisons because they did not exhaustively analyzed their data. However, even with just the 4 comparisons performed (Table 2,3,4,5), both the no resection group data at day 7 and the data at day 30 was each used for 2 comparisons.
The experimental design in the current paper involves multiple groups and multiple timepoint. Use of proper statistical tests should differentiate between effect of the animals, time, tumor status, and error. The author’s choice to not explicitly test some of the combinations does not change the underlying experiment design, the non independence between groups, or the statistical errors that arises when multiple comparisons are made. The generally accepted statistical workflow is that an ANOVA should be perform first, and then followed by post-hoc tests that are corrected for multiple comparisons. Without such corrections, the family-wise error rate will be too high, i.e. more of the detected proteins may be false positive than the authors estimated.
b) (previously point 3c) I have not previously encountered the randomization theory by the authors in their other publication and I do not have the expertise nor time to assess that work. I would assume if corrections for multiple comparisons were performed, the estimated false-positive rate should decrease.
c) (previously point 3d) I would argue all, and not only selected, control groups need to be tested. Even if the controls do not provide the most ‘critical’ comparisons. The statement by the author: “comparison between Day 7 and Day 30 in the control group may have reflected changes in mouse growth” is exactly why that comparison need to be made. Proteomics changes that are dependent on growth/aging should be differentiated from those that are dependent on tumor (resection).
I don’t believe there is any reason why the data should not be fully analyzed. The animal has already been sacrificed and the data collected. A more through analysis can only improve the validity of the conclusions.

Validity of the findings

a) I have some reservations about the main conclusion that urine proteome can be used to assess tumor resection because of the potentially inflated false positive rate, the incomplete analysis of the controls, and only 4 proteins was consistently detected across both time points (after omitting the proteins that were also detected to change in the healthy vs resection 30 day control) (Fig. 6). I believe the first two issues can be addressed with improved data analysis, the last point needs to be addressed in text, e.g. discuss biomarkers vs time points as Reviewer 2 mentioned, highlight the 4 proteins in the discussion, and discuss whether the fold changes were large/consistent.

Additional comments

There are differences in the text of the pdf and docx. The docx file appears to contain more mistakes.

Reviewer 2 ·

Basic reporting

The authors has addressed some of my comments. The scope of research is confined to a really small group of people. The background regarding using urine for biomarker detection is still not firmly established.

Experimental design

This entire study was limited due to short of funding and resources, with only a small number of mice was used to make statistical comparison.

Validity of the findings

Regarding validity of findings, the random grouping model was established in the same research group and the authors referred only their research for making a conclusion that a false-positive rate of 47% means similar condition between complete resection and healthy control. The authors also did not show the data from the healthy group for comparison on day 7 and day 30. The data validity is concerning.

Reviewer 3 ·

Basic reporting

1, I suggest the authors do a thorough proofreading to correct all the typos. For example, line 138: prptides, line 208: defferent, line 309: norma, micce, etc.
2, Line 118-119, please double check the writing here, add “in the healthy control group”, and correct the typo “samll”.

Experimental design

3. The name non resection group is confusing since tissue and muscle were resected from the mice in this group. I suggest the authors consider using tumor-bearing group instead. To be consistent with the names in the title, maybe rename the complete resection group as tumor-resected group. So that you have tumor-resected group, tumor-bearing group, and healthy control group. Please make sure the names are consistent throughout your manuscript.

Validity of the findings

no comment

---

## Round 0.3 · Minor Revisions

Please address remaining issues pointed by the reviewer and amend manuscript accordingly.

Reviewer 3 ·

Basic reporting

The authors have addressed most of my previous comments, I only have a few minor edits this time. Please see below.
Line 67, who are “they”? It is not clear to me.
Lines 125-126, “small section of the tissue and muscle were removed”. I assume this is the healthy control group since you’ve talked about tumor-bearing and tumor-resected groups before this sentence. Please clarify.
For the middle panel of Figure 2, is “recurrence” the right word to use after 30 and 90 days of tissue removal?

Experimental design

no comment

Validity of the findings

no comment

---

## Round 0.4 · accepted · Accept

Thank you very much for addressing the remaining concerns of the reviewer and for amending the manuscript accordingly. I am pleased to inform you that your manuscript is acceptable now.